# Fetal demise and failed antibody therapy during Zika virus infection of pregnant macaques

Diogo M. Magnani [1], Thomas F. Rogers[2], Nicholas J. Maness[3], Nathan D. Grubaugh[2], Nathan Beutler[2], Varian K. Bailey[1], Lucas Gonzalez-Nieto[1], Martin J. Gutman[1], Núria Pedreño-Lopez[1], Jaclyn M. Kwal[1], Michael J. Ricciardi[1], Tereance A. Myers[3], Justin G. Julander[4], Rudolf P. Bohm[3], Margaret H. Gilbert[3], Faith Schiro[3], Pyone P. Aye [3], Robert V. Blair[3], Mauricio A. Martins[1], Kathrine P. Falkenstein[3], Amitinder Kaur[3], Christine L. Curry[5], Esper G. Kallas[6], Ronald C. Desrosiers[1], Pascal J. Goldschmidt-Clermont[7], Stephen S. Whitehead[8], Kristian G. Andersen[2,9,10], Myrna C. Bonaldo[11], Andrew A. Lackner[3], Antonito T. Panganiban[3], Dennis R. Burton[2,12] & David I. Watkins[1]

Zika virus (ZIKV) infection of pregnant women is associated with pathologic complications of fetal development. Here, we infect pregnant rhesus macaques (*Macaca mulatta*) with a minimally passaged ZIKV isolate from Rio de Janeiro, where a high rate of fetal development complications was observed. The infection of pregnant macaques with this virus results in maternal viremia, virus crossing into the amniotic fluid (AF), and in utero fetal deaths. We also treated three additional ZIKV-infected pregnant macaques with a cocktail of ZIKV-neutralizing human monoclonal antibodies (nmAbs) at peak viremia. While the nmAbs can be effective in clearing the virus from the maternal sera of treated monkeys, it is not sufficient to clear ZIKV from AF. Our report suggests that ZIKV from Brazil causes fetal demise in non-human primates (NHPs) without additional mutations or confounding co-factors. Treatment with a neutralizing anti-ZIKV nmAb cocktail is insufficient to fully stop vertical transmission.

[1] Department of Pathology, University of Miami Leonard M. Miller School of Medicine, 1951 NW 7th Ave Room 2340, Miami, FL 33136, USA. [2] Department of Immunology and Microbiology, The Scripps Research Institute, 3215 Merryfield Row Immunology 308, San Diego, CA 92121, USA. [3] Tulane National Primate Research Center, 18703 Three Rivers Road, Covington, LA 70433, USA. [4] Institute for Antiviral Research, Utah State University, 5600 Old Main Hill, Logan, UT 84322-5600, USA. [5] Department of Obstetrics and Gynecology, University of Miami Leonard M. Miller School of Medicine, CRB 11th floor, Miami, FL 33136, USA. [6] Division of Clinical Immunology and Allergy, School of Medicine, University of São Paulo, Av. Dr. Arnaldo 455, Terceiro andar, São Paulo, SP 01246-903, Brazil. [7] Department of Medicine, University of Miami Leonard M. Miller School of Medicine, 1600 NW 10th Ave #1140, Miami, FL, USA. [8] Laboratory of Infectious Diseases, National Institute of Allergy and Infectious Diseases, National Institutes of Health, Bldg 33, Room 3W10A, 33 North Drive, MSC 3210, Bethesda, MD 20892-3210, USA. [9] Scripps Translational Science Institute, 10550 North Torrey Pines Road, La Jolla, CA 92037, USA. [10] Department of Integrative Structural and Computational Biology, The Scripps Research Institute, 10550 North Torrey Pines Road, 92037 La Jolla, CA, USA. [11] Laboratório de Biologia Molecular de Flavívirus, Instituto Oswaldo Cruz, Fiocruz, Avenida Brasil, 4365, Manguinhos, Rio de Janeiro, RJ 21040-360, Brazil. [12] Ragon Institute, Harvard Medical School, 400 Technology Square, Cambridge, Boston, MA 02139, USA. These authors contributed equally: Diogo M. Magnani, Thomas F. Rogers. Deceased: Andrew A. Lackner. Correspondence and requests for materials should be addressed to D.R.B. (email: burton@scripps.edu) or to D.I.W. (email: dwatkins@med.miami.edu)

ZIKV has emerged in Brazil as a significant new threat to human health and has been linked to neurological complications including Guillain–Barré syndrome and fetal developmental defects[1–5]. The birth defects defined as congenital Zika syndrome (CZS) include, in addition to microcephaly, health complications involving vision, hearing, seizures, and motor disabilities[1, 5]. In Rio de Janeiro, 42% of fetuses of pregnant women infected with ZIKV had abnormalities[2]. Varying incidence of CZS have been reported[2, 4, 6] with considerable controversy surrounding whether ZIKV alone is sufficient to cause fetal developmental problems[7].

Non-human primate (NHP) species are considered the best animal models of human pregnancy and placentation[8]. While several groups have reported severe pathology of the ZIKV-infected fetus, it has been difficult to demonstrate fetal demise in infected pregnant macaques[9–11]. Fetal brain lesions were reported in NHPs after infection of a third trimester pigtail macaque with a high dose ($5 \times 10^7$ pfu) of a Cambodian ZIKV isolate[12]. Recently, Mohr et al.[13] have reported fetal demise following ZIKV PRVABC59 infection in a rhesus macaque. Nonetheless, this animal also had a bacterial infection, and it remains unclear if ZIKV infection alone is sufficient to cause demise of the rhesus fetus.

Here we show that the ZIKV Rio U-1/2016 strain, isolated from the urine of a pregnant woman living in a region with high incidence of CZS, causes fetal demise in a NHP host. We have infected 11 rhesus macaques with ZIKV, at different times during the pregnancy. We inoculated the pregnant rhesus macaques with 10,000 plaque forming units (pfu) of a minimally passaged ZIKV isolate from Rio de Janeiro, where a 42% rate of fetal developmental problems was previously observed in pregnant women.

This virus caused lethality in AG129 mice, at lower doses than other common ZIKV isolates. In infected pregnant macaques, it resulted in prolonged maternal viremia and viral crossing into the amniotic fluid (AF) of five pregnant macaques. Importantly, two fetuses of three dams infected in the first trimester died in utero. In one of these fetuses, ZIKV was present in the AF and chorionic fluid (CF) ($>10^5$ copies ml$^{-1}$) at the time of fetal demise. We also treated three additional ZIKV-infected pregnant animals with a cocktail of ZIKV-neutralizing human monoclonal antibodies (nmAbs) at peak viremia. While the nmAbs were effective in clearing the virus from the sera of two of the three treated animals, ZIKV was still detected in the AF of one of the treated animals. Our results indicate that ZIKV Rio U-1/2016 is sufficient for causing fetal demise in the rhesus NHP model of ZIKV-infection. Furthermore, treatment with a potent nmAb cocktail did not prevent vertical transmission. Therapeutic approaches for ZIKV using nmAbs will have to specifically designed for reaching effective concentrations in the fetal compartment.

## Results

**ZIKV Rio U-1/2016 infection of mice and rhesus macaques.** Infection of nonpregnant AG129 mice with the Rio U-1/2016 isolate[14] showed that this virus was lethal after challenge (Supplementary Fig. 1). Furthermore, low doses of this virus were lethal to mice, indicating that this strain might be more pathogenic than the other common ZIKV PRABC59 or Paraiba/2015 isolates (Supplementary Fig. 1). We then used this low-passaged ZIKV isolate to challenge rhesus macaques in different phases of pregnancy (Fig. 1; Supplementary Table 1). Pregnancy in the rhesus monkey is divided into 55-day trimesters, which are

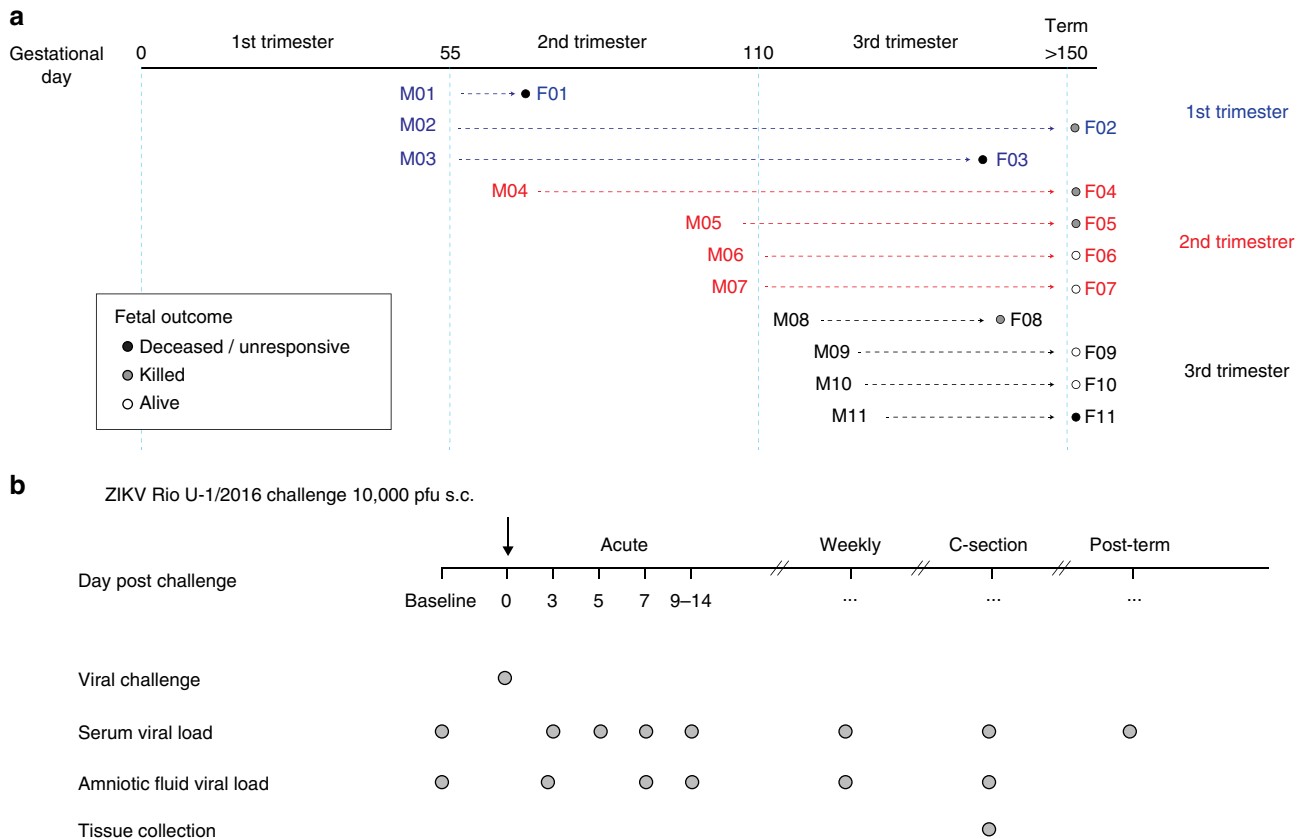

**Fig. 1** Pregnant NHP study design. **a** We challenged 11 rhesus macaques with 10,000 pfu of ZIKV Rio U-1/2016 during the first (blue), second (red), and third (black) trimester (55 days each) of pregnancy. Fetal outcomes; deceased/unresponsive, killed, or alive are indicated by black, gray, and white, circles, respectively. **b** Dam and fetal samples were collected at the indicated time points

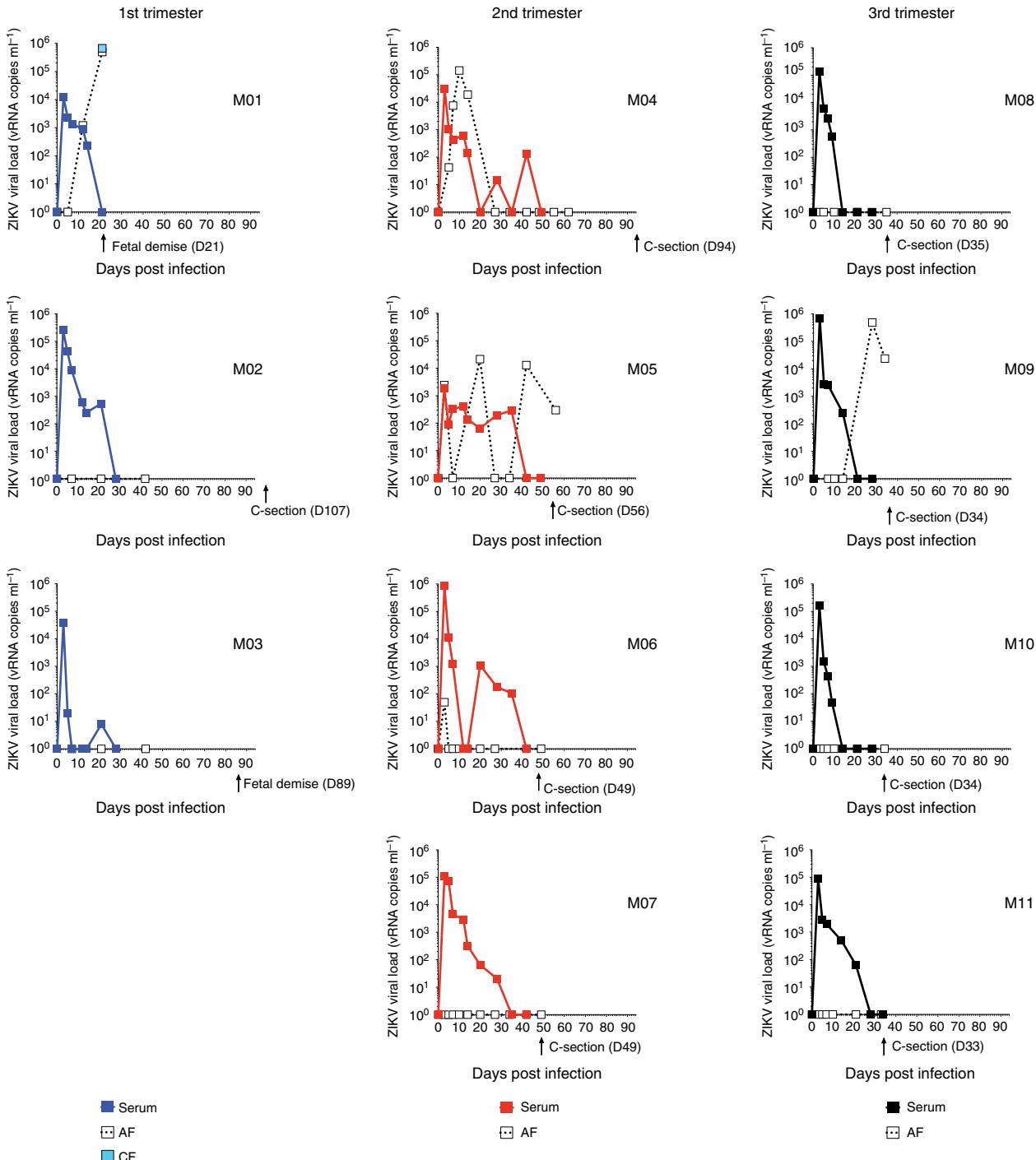

**Fig. 2** Pregnant macaques have extended viremia and transmit virus to the fetus. Blood and AF were collected at the indicated time points. At time of C-section or fetal demise, CF was also harvested. Serum (filled square), AF (open square), CF (light blue square) viral loads for 11 macaques in the first (blue), second (red), or third (black) trimester of pregnancy are shown. Date of C-section is indicated by an arrow

equivalent in developmental landmarks to the trimesters of human pregnancy[15]. Ultrasound-guided amniocentesis was performed, and sera were collected at the indicated time points (Figs. 1 and 2).

**Fetal outcomes of ZIKV-infected pregnant macaques.** The pregnant macaques had viremia beyond day 7 post infection, which contrasts to non-pregnant macaques in prior studies[16], in which viremia was cleared by day 7 (Fig. 2). The first animals to

be born after ZIKV infection (F08, F09, F10, and F11) were delivered ~7 weeks post challenge of the third trimester animals. One of these infants, F11, was not responsive after delivery and was killed. We looked for the presence of viral nucleic acid by qRT-PCR in F11's necropsied tissues (placenta, umbilical cord, and brain), but could not detect virus. Despite having the most prolonged viremia of the third trimester infected animals, the AF was negative for ZIKV RNA at the time of the C-section. The other infants from dams infected during the third trimester had no apparent deficiencies.

Three of the four dams infected during the second trimester experienced transfer of ZIKV to the AF. In two of these dams, M04 and M05, viral levels in the AF were higher than observed in the sera of the dams. Furthermore, M05 virus was observed in the AF when viral RNA was not detected in the maternal serum. In spite of the presence of viral RNA in the AF, these pregnancies were carried to term with no apparent complications.

Notably, two of the three dams challenged during the first trimester of pregnancy experienced fetal demise. Twenty-one days after inoculation of one of the first trimester macaques, ultrasound examination revealed that the fetus from dam M01 was no longer viable. The associated fetal fluids were collected by C-section and were analyzed for the presence of virus. ZIKV was present in both the CF (~660,000 viral RNA copies ml$^{-1}$) and AF (~500,000 viral RNA copies ml$^{-1}$) at a time when there was no virus detected in the serum of the dam at 21 days post-infection (Fig. 2). Similarly, dam M03 experienced fetal loss 89 days after infection. Thus, two of the three first trimester dams challenged with ZIKV experienced fetal loss. Unfortunately, the tissues from the lost fetuses were extensively autolyzed at time of collection, which limited our ability to detect virus or evaluate the tissue pathology. We were unable to detect virus in the tested fetal tissues (lymph nodes, brain). Using a comparable study design, Mohr et al.[13] have recently reported ocular pathology, prolonged viral replication in maternal and fetal macaque tissues, and fetal demise following ZIKV PRVABC59 infection in a rhesus macaque. However, a concomitant bacterial infection during that pregnancy obfuscated the association of the adverse pregnancy outcomes with the ZIKV challenge.

The highest risk of severe human CZS occurs following ZIKV infection in the first trimester of pregnancy[2, 4]. Similarly, two of the three pregnant macaques challenged with ZIKV in the first trimester resulted in fetal demise. The observed fetal loss in the first trimester of pregnancy is consistent with another report of ZIKV-associated fetal demise in macaques[13]. Importantly, fetal loss has been only rarely observed in studies involving dams in the first trimester at the Tulane National Primate Research Center (TNPRC). Even after cytomegalovirus challenge of first trimester pregnant rhesus macaques in a prior study, no fetal loss was observed in immunocompetent animals ($n = 3$), despite repeated amniocentesis and detection of virus in the AF[17]. Likewise, the fetuses from all five control animals of a malaria infection study during pregnancy were brought to term without complications[18]. Furthermore, no spontaneous miscarriages were observed in a group of 30 pregnant rhesus monkeys that were monitored by ultrasound[19]. The overall miscarriage rate has been previously estimated to be 4–7% of known pregnancies in the TNPRC rhesus breeding colony[19]. Of note, these aforementioned studies had experimental designs similar to the current ZIKV challenge, including procedures and collection schedules, using macaques from the same colony. Thus, fetal death is a rare outcome for healthy pregnant macaques at the TNPRC. Even though our study is limited by the lack of a control group, the absence of fetal demise in similar studies with repeated amniocentesis suggests that ZIKV causes fetal demise in macaques.

**Treatment of ZIKV-infected pregnant macaques**. We next attempted to block fetal infections with a cocktail of ZIKV-nmAbs. We recently demonstrated that this SMZAb cocktail prevented ZIKV infection of non-pregnant adult rhesus macaques[16]. As a strategy to treat ZIKV-infected pregnant mothers and their fetuses, we delivered our cocktail (each nmAb delivered at 20 mg kg$^{-1}$) to three macaques during peak viremia (Fig. 3a) at day 3 post-infection, at a time when pregnant women might first notice symptoms of ZIKV. We engineered these nmAbs with Fc LALA mutations[20] that abrogate Fc-gamma receptor binding to eliminate potential therapy-mediated antibody-dependent enhancement. We monitored viral loads in the mother's serum and the AF. We challenged one additional pregnant animal as a contemporaneous untreated control (M15). The SMZAb cocktail attained a peak concentration of 600 μg ml$^{-1}$ of sera and 12 μg ml$^{-1}$ of AF (Fig. 3b). The ZIKV-challenged animals had peak viremias of $10^5$–$10^6$ vRNA copies ml$^{-1}$ of sera on day 3 post infection (Fig. 3c). The administration of the nmAb cocktail completely cleared viremia in the blood of two of three treated animals by day 5 (Fig. 3c). Surprisingly, AF from one of these animals (M13) was positive for ZIKV at day 16, post infection, more than 9 days after viremia in the maternal blood was cleared.

Our mAb cocktail failed to control viremia in one of the treated pregnant dams and virus crossed the placenta in one of the other two treated macaques. Since these third trimester dams were close to delivery, our protocol called for the termination of the experiment at 14–21 days post treatment. Given that these treated macaques were late stage third trimester dams, we would be unlikely to see any fetal complications. Nevertheless, we attempted to amplify ZIKV from all four fetuses but could not amplify virus from any of the three treated fetuses. The fetus from the control animal (M15) was infected, and we were indeed able to detect ZIKV RNA in the AF ($1.7 \times 10^5$ vRNA copies ml$^{-1}$), fetal lymph node ($6.1 \times 10^4$ vRNA copies mg$^{-1}$), and fetal serum ($2.4 \times 10^3$ vRNA copies ml$^{-1}$). In conclusion, the experiment shows clearly that a nmAb cocktail that protects adults from infection[16] did not abrogate viremia in one of three macaques. Furthermore, in the two macaques in which viral replication was curtailed, virus was seen in the AF at a time when there was no detectable virus in the maternal circulation. Thus, mAb treatment following ZIKV infection in pregnant subjects is unlikely to prevent transfer of virus to treated fetuses.

**Viral sequences**. To determine if any intrahost ZIKV genetic variants were associated with acute viremia, in utero transmission, or fetal demise, we sequenced the viral inoculum, virus in the dams' serum ($n = 11$), and virus from the AF ($n = 4$) and CF ($n = 1$; Fig. 4; Supplementary Table 2). Consistent with other studies[21, 22] using non-pregnant macaques, we found that acute viremia in pregnant macaques was not associated with significant ZIKV genetic change (Fig. 4a,b). In fact, we only detected one nonsynonymous intrahost variant at >25% frequency in serum from 11 pregnant macaques (animal M06, C6472T 40% frequency), and it was also present in the inoculum at a 2% minor allele frequency (Supplementary Table 2). We only detected five ZIKV variants of >25% frequency in the AF from four animals (Fig. 4a,b), none of which were detected in the inoculum (Supplementary Table 2). However, most of the variants were synonymous (3/5) and none were convergent (i.e., found in more than one animal). These data suggest that random genetic bottlenecks, rather than selective barriers, were the strongest evolutionary forces during in utero transmission. Moreover, no ZIKV variants were associated with fetal demise in our cohort. Thus, our ZIKV isolate (Rio U-1/2016[3]) does not require any additional mutations to infect pregnant primates, traverse the placenta, or cause fetal disease. Due to our limited sampling, however, it is possible that other ZIKV genetic factors could affect fetal infection or disease.

We also investigated if escape mutations could explain viremia in the nmAb-treated animal M12 (Fig. 4c). Our sequencing data from the serum of this viremic animal did not reveal any de novo

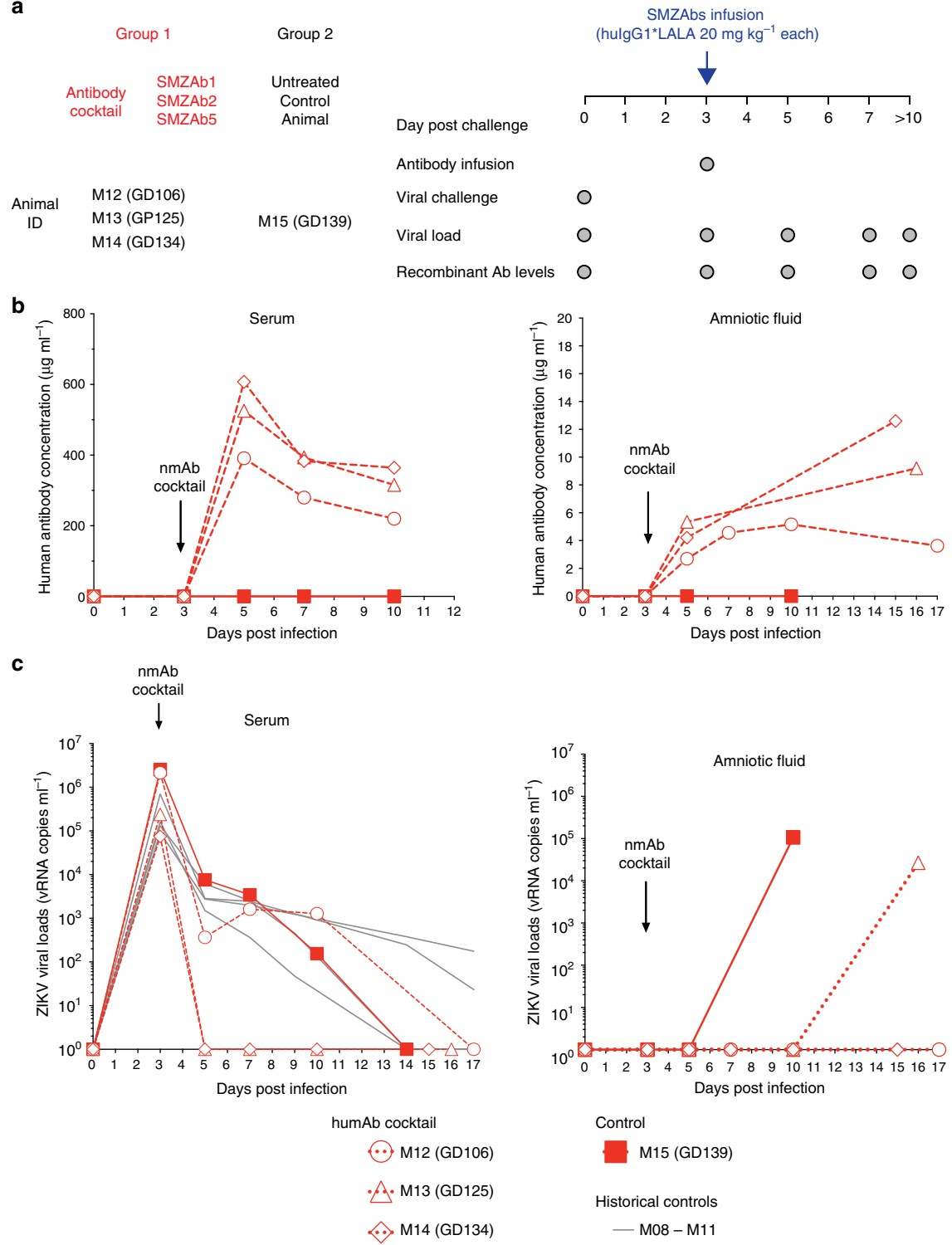

**Fig. 3** Therapeutic administration of the SMZAb cocktail to ZIKV-infected pregnant Indian-origin rhesus macaques blocks maternal viremia but does not prevent transmission of virus to fetus. **a** Treatment study design. Four pregnant rhesus macaques were challenged with 10,000 PFU of ZIKV Rio U-1/2016. On day 3 post challenge, we administered a cocktail containing the three nmAbs, SMZAbs 1, 2, and 5 at a dose of 20 mg kg$^{-1}$ each to three rhesus macaques (Group 1). An untreated animal was used as a control (Group 2). Four macaques infected in the third semester of pregnancy (M8–M11) were used as historical controls. **b** Serum and AF were collected at the indicated time points for viral load and IgG measurements. **c** Serum viral loads for SMZAb-treated (open symbols) and control (solid square) macaques

escape mutations in the ZIKV envelope coding region. Thus, replication of the viral swarm of the wild-type virus in maternal blood and the AF can occur despite treatment of pregnant animals with a potent ZIKV nmAb cocktail.

## Discussion

We have shown that ZIKV infection causes fetal demise in an experimental NHP model. We used a relatively low challenge dose[23] of a minimally passaged virus stock from a region where

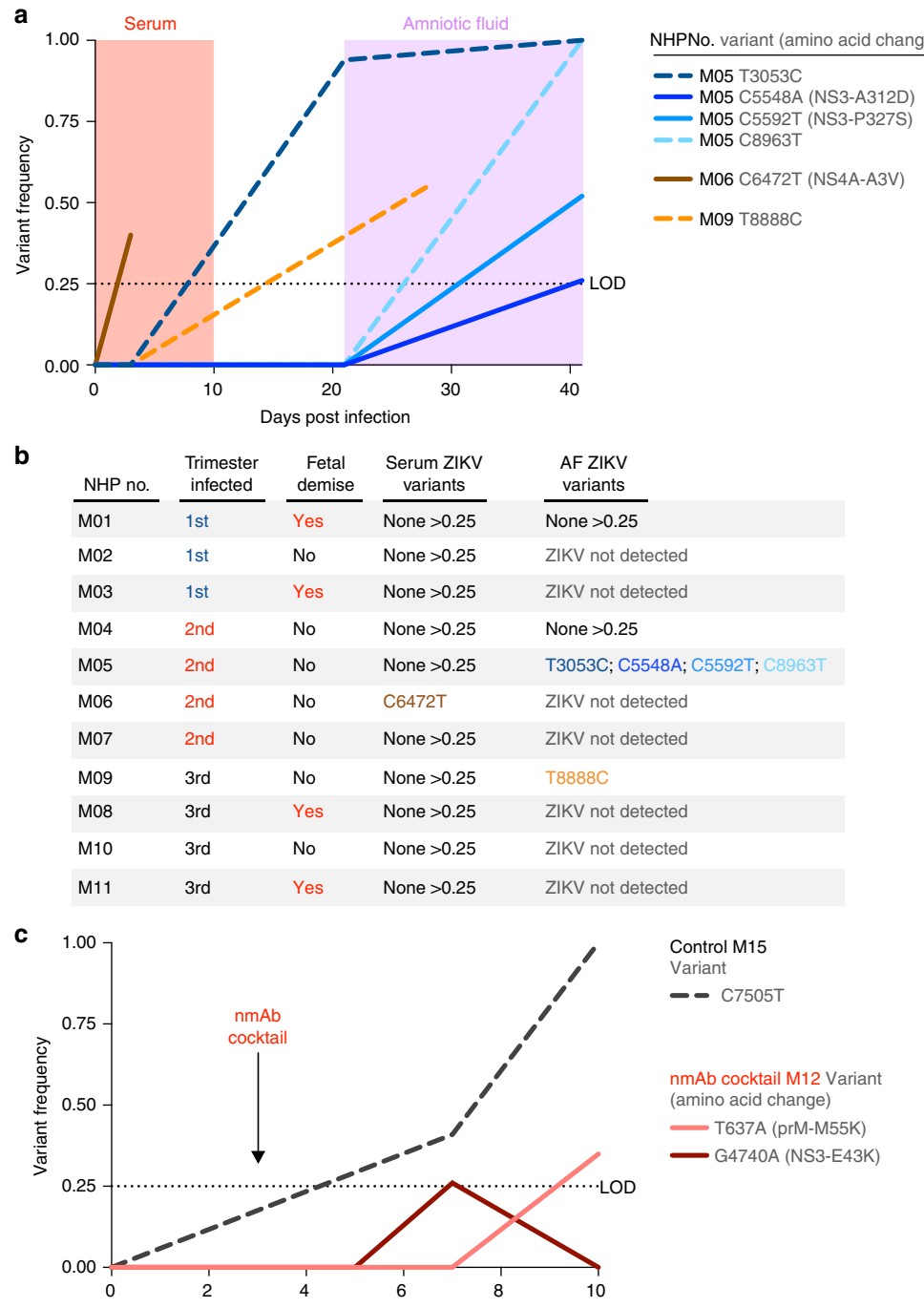

**Fig. 4** De novo mutations in the consensus sequences are not required for in utero transmission, fetal demise, or antibody escape during pregnancy. ZIKV from serum and AF was sequenced using a PCR amplicon-based approach. A very conservative 0.25 frequency cut-off was used detect high frequency intrahost variants. **a** ZIKV intrahost variant frequencies were plotted by the sample source (serum or AF) and days post infection. All blue lines indicate different ZIKV variants sequenced from the same macaque (M04). **b** ZIKV intrahost variants >0.25 corresponding to a specific macaque, trimester of infection, and fetal demise are also shown in a table for clarity. **c** Following nmAb cocktail treatment, one pregnant NHP (M12) failed to control viral replication in the blood. Intrahost ZIKV variants sequenced from the serum of M12 and a no-antibody control were plotted by frequency and days post infection. Variants shown as solid lines represent nonsynonymous mutations and variants shown as dashed lines represent synonymous. The dotted lines at 0.25 frequency represent the limits of detection (LOD)

42% of ZIKV-infected pregnancies resulted in CZS[2]. Infection of pregnant macaques with this stock caused extended viremia in the pregnant dams, ZIKV transfer to the AF in almost half of the pregnancies, and fetal demise of three fetuses. The CDC clinical guidance for the management of pregnant ZIKV patients is to perform PCR in maternal samples[24]. However, our observation of

ZIKV RNA in the AF during aviremic periods in the mother in four animals suggests that ZIKV RNA testing in maternal blood might not be a reliable indicator of fetal infection. No specific ZIKV mutations were required for in utero transmission or fetal demise. These results suggest that wild type circulating ZIKV can cause fetal demise without further adaptation. A recent study has

shown that passive administration of a nmAb reduced, but did not prevent, ZIKV placental viremia in a fetal infection model of pregnant mice[25]. Of note, in contrast to rodents, the placentation of NHPs have marked developmental similarities to humans, in terms of placenta type, extent of cell invasiveness, and cell layer composition[26]. It remains unclear why the passive administration of a nmAb cocktail failed to block ZIKV transmission to the NHP fetal compartment. It is possible that the levels of nmAbs that reached this site (~10% of maternal serum levels) were insufficient to prevent replication.

ZIKV infection during the first two trimesters of pregnancy holds the highest risk of severe fetal disease for ZIKV-infected mothers[2, 4]. Thus, effective therapies of any nature will have to reach the fetus during this early period, in effective amounts and without toxicity. While fetal immunity is acquired through placental translocation of maternal IgG[26], most of the IgG transport occurs in the last trimester of the pregnancy. It is unlikely, then, that our nmAb cocktail would reach the fetal compartment at earlier points in the pregnancy. However, it may be possible to enhance the rate of transplacental mAb transport. Further modification of our nmAbs with mutations that facilitate FcRn-mediated functions[27] may be necessary for the success of this approach.

## Methods

**Animal experiments**. The 15 Indian origin rhesus macaques (*Macaca mulatta*) utilized in this study were housed at the TNPRC. The TNPRC is fully accredited by AAALAC International (Association for the Assessment and Accreditation of Laboratory Animal Care), Animal Welfare Assurance No. A3180-01. Animals were cared for in accordance with the NRC *Guide for the Care and Use of Laboratory Animals* and the Animal Welfare Act Animal experiments were approved by the Institutional Animal Care and Use Committee of Tulane University (protocol P0336). Pregnant macaques were selected based on availability and were not randomized during assignment. No blinding was done.

Male and female AG129 mice lacking interferon type I and type II receptors were produced from an in-house colony. The original five breeding pairs of AG129 mice were obtained in 2008 from Dr. Robert Schreiber, Washington University Medical School (Saint Louis, MO). Groups of five mice, from 5 to 9 weeks of age and an average weight of $23.3 \pm 3.4$ g, were challenged s.c. in the inguinal fold with various dilutions of ZIKV. Mice were observed twice-daily throughout the study for signs of disease and mortality. Mice were humanely killed when neurological manifestations of virus infection, including loss of motility, lateral recumbancy, weight loss >30% or the inability of the mice to access food and water. The death date for animals that met the humane killing criteria was recorded as the day after they were killed. All work was done in the AAALAC-accredited Laboratory Animal Research Center at Utah State University under institutionally approved protocols.

Fetal tissues were collected in necropsies performed after fetal demise (C-sections). The materials sampled depended on the status of tissue preservation, but included brain, optic nerves, lymph nodes, placenta, cord-blood, bone marrow, spinal cord, spleen, liver, and kidney.

**Viruses and challenge**. ZIKV strain Rio U-1/2016 was isolated in Rio de Janeiro, Brazil in 2016 (KU926309). Viral challenge stocks were prepared by propagating the virus in Vero cells Vero (American Type Culture Collection CCL81TM) for two passages post-virus isolation[14]. The stocks were quantitated by viral plaque assay. The viral stocks were diluted in Leibovitz's L-15 and SPG media as described previously[28]. All animals were challenged via the subcutaneous route, with 10,000 PFU.

**Passive antibody administration**. A cocktail of three anti-ZIKV mAbs was produced by combining SMZAb1, SMZAb2, and SMZAb5[16]. Each mAb was delivered at a dose of 20 mg kg$^{-1}$. The macaques were separated in two groups, as follows: Group 1 (SMZAb cocktail, $n = 3$); Group 2 (untreated, $n = 1$). The nmAb cocktail was prepared in saline and administered intravenously into each animal. All animals were infused with the nmAb cocktail 3 days post challenge. After the challenge, we collected serum and AF at the indicated time points to measure viremia, nmAb levels.

**Measurement of viral RNA load (qRT-PCR)**. Quantitative realtime PCR (qRT-PCR) was used for the measurement of viral loads, based on a previously validated assay[29, 30]. In brief, RNA was extracted from 140 to 1000 μl of frozen fluids, depending on availability, using the QIAamp Viral RNA Mini Kit or QIAamp Circulating Nucleic Acid kit (Qiagen, Hilden, Germany). The total nucleic acid was eluted in two centrifugation steps with 40 μl of Buffer AVE each. A qRT-PCR reaction was then carried out with 20 μl of samples and 10 μl of primer, probes, and TaqMan Fast Virus 1-Step Master Mix (Applied Biosystems, Foster City, CA). We used pre-combined probe and primers (500 nM primers and 250 nM probe; IDT Technologies, Coralville, IA). The primer and probe sequences were designed to match the sequences of the Brazilian ZIKV isolate KU321639 and were as follows: primer 1 5′TTGAAGAGGCTGCCAGC3′; primer 2 5′CCCACTGAACCCCATC-TATTG3′; probe 5′TGAGACCCAGTGATGGCTTGATTGC3′. The probe was double-quenched (ZEN/Iowa Black FQ) and labeled with the FAM dye (IDT Technologies, Coralville, IA). Tenfold serial dilutions of a 401 bp in vitro RNA transcript encoding the ZIKV capsid gene (KU321639) starting at approximately $5 \times 10^5$ RNA copies μl$^{-1}$ were used as standards. Results were reported as the median equivalent viral RNA genomes per ml. The limit of detection was between 12 and 90 viral RNA copies ml$^{-1}$, depending on the extracted volumes. ZIKV-positive and ZIKV-negative samples were included in every run. Viral load data was analyzed with QuantStudio Real-Time PCR Software v1.3 and graphed using GraphPad Prism 7.0a.

**Zika virus genetic analysis**. RNA was extracted from the samples by the same methods as described in the qRT-PCR section. Deep sequencing was performed using a PCR amplicon-based approach[31, 32]. Briefly, cDNA was reverse transcribed from 5 μl of RNA using SuperScript IV (Invitrogen). ZIKV cDNA (2.5 μl per reaction) was amplified in 35×400 bp fragments from two multiplexed PCR reactions using Q5 DNA High-fidelity Polymerase (New England Biolabs). The amplified ZIKV cDNA fragments (50 ng) were prepared for sequencing using the Kapa Hyper prep kit (Kapa Biosystems) and SureSelect XT2 indexes (Agilent). Agencourt AMPure XP beads (Beckman Coulter) were used for all purification steps. Paired-end 251 nt reads were generated on the MiSeq using the V2 500 cycle kit (Illumina).

Demultiplexing was performed by the Illumina instrument. The primer sequences were removed from the reads and bases with Phred quality scores <20 were removed by Trimmomatic[33]. The reads were then aligned to the complete genome of a ZIKV isolate from the Dominican Republic, 2016 (GenBank: KU853012) using Novoalign v3.04.04 (www.novocraft.com). Samtools was used to sort the aligned BAM files[34]. The consensus and intrahost variants >0.25 frequency were called using Geneious v9.1.5[35] before generating consensus sequences. Calling consensus nucleotides at each position required at least ×25 coverage and variant calling required at least ×100 coverage. The consensus sequence for the input Zika virus (Rio U-1/2016) was 100% identical to the previously published sequence (GenBank KU926309). A conservative 0.25 frequency cut-off was used to (1) help account for variations in input ZIKV concentrations and (2) to focus on mutations that may be under strong selective pressures. The sequencing statistics are presented in Supplementary Table 2 and the data are available in the NCBI BioProject collection (PRJNA413046).

**In vivo human antibody quantitation by ELISA**. The presence of the recombinant nmAbs in sera was quantitated with an ELISA specific for human Abs. In brief, 96-well ELISA plates were coated overnight with 5 μg ml$^{-1}$ of the monkey Ab-adsorbed goat anti-human IgG (Southern Biotech, 2049-01) diluted in phosphate-buffered saline (PBS). Each plate was washed with PBS-Tween20 and the wells were blocked with 5% non-fat dry milk in PBS for 1 h at 37 °C. Subsequently, the plate was washed with PBS and serum samples were added to designated wells. After 1 h incubation at 37 °C, the plate was washed, and detection was carried out using a HRP-conjugated goat anti-human IgG (Southern Biotech, 2045-05), which was added to all wells at a dilution of 1:10,000. Following a 1 h incubation at 37 °C, the plate was washed with PBS-Tween20 and developed with TMB substrate (Millipore) at room temperature for 3–4 min. Reaction was then stopped with TMB stop solution, and absorbance was read at 450 nm.

**Data availability**. The data sets generated during and/or analyzed during the current study are available from the corresponding author on reasonable request. The ZIKV sequencing data is available in the NCBI BioProject collection under accession code PRJNA413046.

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

## Acknowledgements

This work was supported by the Bill and Melinda Gates Foundation (OPP1152818), the National Institutes of Health (NIH) grant 4P01AI094420-05, the Wallace H. Coulter Center for Translational Research at the University of Miami, the Miami Clinical and Translational Science Institute (CTSI), the Jacobson Jewish Community Foundation and Clarence Wolf Jr & Alma B Wolf Foundation, and the Intramural Research Program of the National Institute of Allergy and Infectious Diseases, NIH. This work was in part supported by NIH grant P51 OD011104 to the TNPRC, the NIH grant P30AI073961 to the Miami Center for AIDS Research, as well as the Defense Advanced Research Projects Agency (DARPA) Autonomous Diagnostics to Enable Prevention and Therapeutics: Prophylactic Options to Environmental and Contagious Threats (ADEPT-PROTECT) program (W31P4Q-13-1-0011). The funders had no role in the study design, data collection and analysis, decision to publish, or preparation of the manuscript.

## Author contributions

D.M.M., T.F.R., N.J.M., N.D.G., J.M.K., C.L.C., E.G.K., R.C.D., S.S.W., P.G., R.P.B., M.H.G., M.C.B., A.A.L., A.T.P., D.R.B., and D.I.W., planned the studies. D.M.M., N.J.M., T.F.R., N.D.G., V.K.B., L.G.-N., M.J.G., N.P.-L., N.B., M.J.R., J.G.J., R.P.B., M.H.G., F.S., P.P.A., R.V.B., A.M., A.K., K.P.F., M.A.M., and S.S.W. conducted the experiments. M.C.B. provided reagents. D.M.M., N.J.M., T.F.R., N.D.G., R.C.D., S.S.W., A.T.P., K.G.A, and D.I.W., interpreted the studies. D.M.M., N.D.G., and D.I.W. wrote the first draft. J.M.K., T.F.R., P.G., S.S.W., D.R.B., A.A.L., K.G.A., A.T.P., and D.I.W obtained funding. All authors reviewed, edited, and approved the paper.

## Additional information

**Competing interests:** D.R.B. and T.F.R. are inventors on a patent application submitted by The Scripps Research Institute that covers anti-Zika monoclonal antibodies. The remaining authors declare no competing interests.

