## [Peer Review File · Nature Communications]

Reviewers' comments:

Reviewer #1 (Remarks to the Author):

In the manuscript entitled "Fetal demise and failed antibody therapy during Zika virus infection of pregnant macaques" by Magnani et al. describes the effect of low dose subcutaneous ZIKV challenge on pregnant rhesus macaques to explore the association between ZIKV infection and the outcome of fetal demise. They observe viral replication in all dams and fetal demise in 3 out of 11 pregnancies. In addition, the authors evaluate monoclonal antibody (mAb) therapy during pregnancy. A cocktail of 3 neutralizing mAb was delivered 3 days after subcutaneous ZIKV challenge. A reduction in Zika RNA was observed in 2/3 animals. Cross-placental detection of viral RNA was observed in one macaque. Finally, the authors looked for evidence of viral escape from mAb, but observed no evidence of mAb escape.

In this revised manuscript version, the authors have made numerous modifications that address many of the reviewers' previous concerns. This study presents novel and important information for the field. However, there are several concerns that dampen overall enthusiasm. We have several suggestions to improve the manuscript.

Major concerns

- 1) A recently-published study by Mohr et al. (PLOS One 2018) describes fetal demise following experimental ZIKV infection of a single first-trimester pregnant rhesus macaque. The findings from this study should be discussed in the manuscript.
- 2) In addition to the ZIKV strain used in the monkey studies (ZIKV Rio U-1/2016), Fig. S1A-B shows mouse data for two additional strains (PRVABC59 V2933 and Paraiba/2015). It is not clear to this reviewer what the purpose is of the data describing the additional ZIKV strains. A clarification of these data and their relationship to the subsequent data should be included. Additional information needs to be added to this figure including the units of virus dose and a description of the graphs displayed (Kaplan-Meier?).
- 3) Is it possible to include the third trimester-infected monkeys M09-M11 as additional controls for the antibody study? These controls would strengthen the conclusion that the nmAb was responsible for clearing the virus. Without a comparison for typical duration of maternal viremia in a third trimester infection, it's hard to make this claim.

Minor concerns

- 4) A number of passages need to be edited and corrected. A thorough edit of the manuscript is suggested.
- 5) Please remove sensationalized language such as "Interestingly" and "Surprisingly", for example on lines 101 and 107. We believe that these findings have been reported in clinical studies of Zika virus.
- 6) The reviewers appreciate the limitation of available pregnant macaques for the mAb study. However, the limitations of evaluating the mAb in third-trimester dams should be addressed in more depth in the manuscript.
- 7) Lines 48, 50, 51: the term "animals" should be clarified to specifically mention monkeys/macaques as the abstract discusses both murine and macaque studies. As it presently reads, it could be easily misunderstood.
- 8) Line 92: as this sentence is currently written, the comparison that is being made is unclear. This reviewer suggests "The pregnant macaques had viremia beyond day 7 post infection, which contrasts to non-pregnant macaques in prior studies in which viremia was cleared by day 7.
- 9) There seem to be some editing errors in Figure 2. The Figure refers to a 1st trimester monkey, KP33, which seems to actually be M03 from the text. M08 should be placed before M09.

Responses to the reviewer comments

Nature Communications submission NCOMMS-18-01836-T

Responses are highlighted in **yellow**. Because of the extensive revision, we have not highlighted all changes in the manuscript.

REVIEWER'S COMMENTS.

Reviewers' comments:

Reviewer #1 (Remarks to the Author):

In the manuscript entitled "Fetal demise and failed antibody therapy during Zika virus infection of pregnant macaques" by Magnani et al. describes the effect of low dose subcutaneous ZIKV challenge on pregnant rhesus macaques to explore the association between ZIKV infection and the outcome of fetal demise. They observe viral replication in all dams and fetal demise in 3 out of 11 pregnancies. In addition, the authors evaluate monoclonal antibody (mAb) therapy during pregnancy. A cocktail of 3 neutralizing mAb was delivered 3 days after subcutaneous ZIKV challenge. A reduction in Zika RNA was observed in 2/3 animals. Cross-placental detection of viral RNA was observed in one macaque. Finally, the authors looked for evidence of viral escape from mAb, but observed no evidence of mAb escape.

In this revised manuscript version, the authors have made numerous modifications that address many of the reviewers' previous concerns. This study presents novel and important information for the field. However, there are several concerns that dampen overall enthusiasm. We have several suggestions to improve the manuscript.

Major concerns

1) A recently-published study by Mohr et al. (PLOS One 2018) describes fetal demise following experimental ZIKV infection of a single first-trimester pregnant rhesus macaque. The findings from this study should be discussed in the manuscript.

We have now discussed the indicated reference in the text:

"Using a comparable study design, Mohr et al. have recently reported ocular pathology, prolonged viral replication in maternal and fetal macaque tissues, and fetal demise following ZIKV PRVABC59 infection in a rhesus macaque. However, a concomitant bacterial infection during that pregnancy obfuscated the association of the adverse pregnancy outcomes with the ZIKV challenge."

2) In addition to the ZIKV strain used in the monkey studies (ZIKV Rio U-1/2016), Fig. S1A-B shows mouse data for two additional strains (PRVABC59 V2933 and Paraiba/2015). It is not clear to this reviewer what the purpose is of the data describing the additional ZIKV strains. A clarification of these data and their relationship to the subsequent data should be included. Additional information needs to be added to this figure including the units of virus dose and a description of the graphs displayed (Kaplan-Meier?).

We have included the description requested in the graph caption. The text now includes:
"Furthermore, low doses of this virus were lethal to mice, indicating that this strain might be more pathogenic than the other common ZIKV PRVABC59 or Paraiba/2015 isolates (Supplementary Fig. 1)."

3) Is it possible to include the third trimester-infected monkeys M09-M11 as additional controls for the antibody study? These controls would strengthen the conclusion that the nmAb was responsible for clearing the virus. Without a comparison for typical duration of maternal viremia in a third trimester infection, it's hard to make this claim.

We have now included the M8-M11 viral load data as historical controls.

Minor concerns

4) A number of passages need to be edited and corrected. A thorough edit of the manuscript is suggested.

We have edited several sections in the manuscript.

5) Please remove sensationalized language such as "Interestingly" and "Surprisingly", for example on lines 101 and 107. We believe that these findings have been reported in clinical studies of Zika virus.

We have reduced the use of these words in the text.

6) The reviewers appreciate the limitation of available pregnant macaques for the mAb study. However, the limitations of evaluating the mAb in third-trimester dams should be addressed in more depth in the manuscript.

We have now included the following discussion in the text:

“ZIKV infection during the first two trimesters of pregnancy holds the highest risk of severe fetal disease for ZIKV-infected mothers. Thus, effective therapies of any nature will have to reach the fetus during this early period, in effective amounts and without toxicity. While fetal immunity is acquired through placental translocation of maternal IgG, most of the IgG transport occurs in the last trimester of the pregnancy. It is unlikely, then, that our nmAb cocktail would reach the fetal compartment at earlier points in the pregnancy. However, it may be possible to enhance the rate of transplacental mAb transport. Further modification of our nmAbs with mutations that facilitate FcRn-mediated functions may be necessary for the success of this approach.”

7) Lines 48, 50, 51: the term "animals" should be clarified to specifically mention monkeys/macaques as the abstract discusses both murine and macaque studies. As it presently reads, it could be easily misunderstood.

We have now used 'macaques' and 'monkeys', as per the reviewer suggestion.

8) Line 92: as this sentence is currently written, the comparison that is being made is unclear. This reviewer suggests "The pregnant macaques had viremia beyond day 7 post infection, which contrasts to non-pregnant macaques in prior studies in which viremia was cleared by day 7.

We have included the suggested sentence in the text.

9) There seem to be some editing errors in Figure 2. The Figure refers to a 1st trimester monkey, KP33, which seems to actually be M03 from the text. M08 should be placed before M09.

A new figure with the corrections was generated.

REVIEWERS' COMMENTS:

Reviewer #1 (Remarks to the Author):

All the issues have been addressed. I have no further comments or concerns. Very nice manuscript.